# Low Physical Performance Could Be Associated with Adverse Health Outcomes over Time: Results from a Cohort of Older Adults

**DOI:** 10.3390/ijerph21030319

**Published:** 2024-03-09

**Authors:** Darlise Rodrigues dos Passos Gomes, Leonardo Pozza Santos, Edgar Ramos Vieira, Andréa Dâmaso Bertoldi, Elaine Tomasi, Flávio Fernando Demarco, Maria Cristina Gonzalez, Simone Farias-Antunez, Renata Moraes Bielemann

**Affiliations:** 1Post-Graduate Program in Food and Nutrition, Federal University of Pelotas, Pelotas 96010-610, RS, Brazil; cristina.gonzalez@pbrc.edu (M.C.G.); rbielemann.fn@ufpel.edu.br (R.M.B.); 2Department of Nutrition, Federal University of Pelotas, Pelotas 96010-610, RS, Brazil; lpozza.fn@ufpel.edu.br; 3Department of Physical Therapy, Florida International University, Miami, FL 33199, USA; evieira@fiu.edu; 4Post-Graduate Program in Epidemiology, Federal University of Pelotas, Pelotas 96020-220, RS, Brazil; andreadamaso.epi@gmail.com (A.D.B.); tomasiet@gmail.com (E.T.); ffdemarco@gmail.com (F.F.D.); 5Post-Graduate Program in Dentistry, Federal University of Pelotas, Pelotas 96015-560, RS, Brazil; 6Department of Health Sciences, Federal University of Santa Catarina, Araranguá 88906-072, SC, Brazil; simone.farias.antunez@ufsc.br

**Keywords:** aging, walk tests, longitudinal study, physical tests, morbidity

## Abstract

A few studies on physical performance (PP) decline among community-dwelling older adults have simultaneously evaluated various outcomes in Brazil. This longitudinal cohort study aimed to verify the association between PP and health outcomes (negative health self-perception—NHSP; consultations with health professionals; disability; falls; and hospitalization) in older Brazilians (N = 476, 68 ± 6.7 years). PP assessments included Gait Speed (GS) and Timed Up and Go (TUG) tests, and changes were evaluated over time (2014 to 2019–2020). The association between the PP and the outcomes was estimated using Poisson’s regression with robust variance. The physical tests were not associated with NSPH or with the number of consultations with health professionals. However, after adjustment (economic level, diet quality, physical activity, multimorbidity, depression, polypharmacy, and BMI), low PP at baseline (TUG and GS) was associated with disability at follow-up. A low TUG performance at baseline was also associated with subsequent falls (PR = 1.57, *p* = 0.007). A decline in GS was associated with hospitalization (PR = 1.86, *p* = 0.033). PP was associated with disability, falls, and hospitalization over a five- to six-year period in older Brazilians. Regular PP assessments should be conducted and low PP should be used as an indicator of the need for preventative measures to avoid poor health outcomes.

## 1. Introduction

The World Health Organization estimates that 70% of all older people worldwide will be living in the developing world by 2025, with approximately 8% in Latin America [1]. The older people correspond to 15.8% of the Brazilian population [2]. Recent data show that, among this age group, there has been significant increase in individuals, including people aged 80 and over, in whom the risk of frailty is much higher [2]. In Brazil, the number of centenarians is 37,800 people [2]. Healthy aging represents a major challenge for all countries that need to optimize health, social participation, and safe environments to improve the quality of life of older adults [1].

Physical performance (PP) is the objective measurement of an individual’s ability to integrate various physiological systems (cardiac, respiratory, neuromuscular) to execute coordinated and efficient movements. It is considered a critical health indicator in older adults due to its importance in assessing mobility and overall physical well-being [3]. Furthermore, PP is also used as a diagnostic component of prevalent geriatric conditions, such as sarcopenia and frailty [4,5].

The decline in PP among older people may be the result of the complex interaction between physiological factors (declines in cardiovascular, musculoskeletal, and neuromuscular systems’ functioning, resulting in the progressive loss of muscle mass and strength; reduced testosterone levels), clinical factors (e.g., depressive symptoms, multimorbidity, polypharmacy), lifestyle (e.g., diet quality), and sociodemographic (e.g., age, sex) [6,7,8,9]. However, the decline in PP can be reduced by intervening on potentially modifiable risk factors. In this sense, numerous interventions regarding physical activity have shown progress in mobility, strength, and/or balance among healthy or community-dwelling older people [10,11].

Among the several aspects related to PP, a recent study showed that only low mobility was a predictor of a higher risk of new hospitalizations [12]. Low PP, assessed using Gait Speed (GS) or Timed Up and Go (TUG) tests, has been consistently associated with adverse health outcomes such as frailty, sarcopenia, disability, cognitive decline, falls, nursing admission, hospitalization, and higher mortality in different communities and settings worldwide [3,6,13,14]. Furthermore, changes in PP levels over time may predict negative outcomes in older adults. The reverse is also true; improved PP (indicating a positive change over time) reduces the risk of negative outcomes in older adults. Shuman et al. (2020) [15] showed that each 0.05 m/s increase in GS resulted in an 11% reduction in falls (IRR = 0.89; 95%CI = 0.84–0.94; *p* < 0.0001); those who improved GS had 61 falls per 1000 person months, while those who had no improvement or declined in GS had 135 falls. Nonetheless, the relationship between PP and important health indicators, such as health self-perception (HSP) and the number of consultations with health professionals, is still rarely studied. Moreover, longitudinal investigations of this association among older Brazilian individuals are scarce.

PP and functional capacity tend to decline with advancing age, resulting in high costs to health systems (in terms of hospital admission and need for long-term care) and society [1]. Individuals with worse PP use primary and secondary healthcare services more intensively, visit emergency rooms more often, and are hospitalized more frequently and for longer periods than those with better PP [16]. Reduced PP can increase the financial burden on an already overloaded health system. Therefore, including the assessment of PP in the health routine of the older adults is an important step. Integrated with the assessment of psychological and social dimensions, it contributes to the early identification of the frailty syndrome and the implementation of timely actions and treatments [5].

Brazil is a country that has a public health system with universal coverage. Despite efforts to organize the line of care for the older people, there is still a lack of planning for the allocation of human and financial resources based on tools such as multidimensional assessment by primary care teams, which allows identification as well as risk stratification of frail older adults to guarantee ‘the right patient in the right place and time’ [17]. Understanding the role of PP in adverse health outcomes in older adults can help guide preventive public policies and the strategic use of limited financial resources within the health systems of developing countries. However, most of the studies have a cross-sectional or experimental design, include specific populations (institutionalized older people, patients with specific pathologies), and adjust for few confounding factors. In addition, few studies of PP among community-dwelling older adults evaluated multiple outcomes simultaneously. Therefore, this study evaluated the association between PP and HSP, the number of consultations, physical disability, falls, and hospitalization among community-dwelling older Brazilians over a period of five to six years through the combination of PP measures at a single point (cutoff point for low PP) and long-term changes (according to clinically significant changes in physical tests). We hypothesize that PP measures are associated with negative health outcomes in a different way, that is, depending on the physical test (VM and/or TUG) and the moment evaluated (single point and/or over time). some associations may be found with certain outcomes but not others, reinforcing the complementarity of physical tests and the importance of systematically including these measures in the routine assessments of older adults [18]. Our results may support the implementation of healthcare and management strategies for older people in Brazil, based on the monitoring of PP measures over time by the national health system—as is already the case in developed countries—as well as preventing adverse health outcomes, which contributes to the healthy aging of society as a whole and to the efficiency of services designed to serve the older population, especially in Latin American countries.

## 2. Materials and Methods

### 2.1. Study Population

The study included participants of the Longitudinal Study of Older Adults Health—an ongoing longitudinal cohort study called “COMO VAI?” Consórcio de Mestrado Orientado para Valorização da Atenção ao Idoso (Master’s Consortium Oriented for the Appreciation of Older Adults Care) that was initiated in 2014. The inclusion criteria were adults who were community-dwelling, older than 60 years (in Brazil, individuals aged ≥60 years are considered older adults), and living in an urban area of the city of Pelotas, RS, Brazil (~324,000 inhabitants; 93% urban area) [2]. Those who were unable to answer the questionnaire due to cognitive impairment and who did not have a caregiver to assist them were excluded, as were those who were unable to perform the PP tests in both the baseline (2014) and follow-up interviews (2019–2020). Thus, participants had to carry out the performance tests at both baseline and at the follow-up to be included.

The sample size was calculated based on the study of frailty and sarcopenia in older people from Pelotas. For the study on frailty, the sample size calculation was estimated as 857 individuals, considering the prevalence of the outcome was 30% with a 95% confidence interval, and four percentage points was considered an acceptable error and the design effect was 1.5, plus 20% for losses and refusals [19]. The sample size calculation was estimated as 1121 individuals, considering the prevalence of sarcopenia was 10% with a 95% confidence interval, and two percentage points was considered an acceptable error and the design effect was 1.10, plus 20% for losses and refusals [20]. The sampling process is described elsewhere [19,20]. In brief, recruitment took place in two stages. Initially, 133 census tracts were randomly selected from the total census sectors from Pelotas, based on data from the 2010 Brazilian Demographic Census [21]. In the second stage, 31 households were systematically selected per sector to enable the identification of at least 12 older adults in each sector based on a prior estimate of 0.43 older adults/household. This process resulted in the identification of 1844 individuals eligible to participate in the study (baseline). The number of older adults recruited in 2014, as well as the follow-up rate, identified deaths, losses, and refusals, are presented in Figure 1.

Household interviews were conducted between January and August 2014 (baseline). A structured questionnaire was used to investigate general aspects related to older adults’ health as well as sociodemographic variables. The PP tests (GS and TUG) and anthropometric measurements were assessed by standardized interviewers. In 2016–2017, a new phase of telephone/home interviews was carried out, in addition to monitoring mortality (PP tests were not evaluated in this wave). Complete home-based follow-ups were performed between September 2019 and March 2020. Interviews were conducted to assess health outcomes and PP was reassessed. However, the third phase of the study was interrupted due to the COVID-19 pandemic. Of the 900 individuals targeted to be interviewed in the 2019–202020 follow-up, 537 were actually included. The deaths that occurred before December 2022 were not verified due to the extension of the pandemic and the overload of health surveillance in the city during this period. This study used data from the first and third interviews of the participants of the “COMO VAI?” study.

### 2.2. Physical Performance Assessments

PP was evaluated using the GS and TUG tests. GS is recognized as the best test to estimate overall health condition [3]. Although walking speed is the main component of the Timed Up and Go (TUG) test, it also assesses balance and strength, predicting the risk of falls [22]. Moreover, these tests are valid to measure the risk of negative outcomes such as physical disability, cognitive decline, falls, institutionalization, and mortality in older adults [3,6,13,14]. Both were performed twice during baseline (2014) and follow-up (2019–2020), and the best performances were used for the analysis. Walking aids were permitted if required; however, no caregiver assistance was permitted. GS was assessed using a stopwatch to record the time needed to walk a 4 m linear path without obstacles at the fastest possible speed without running [6], with a static start and stop. The speed was calculated in m/s. TUG performance was assessed as the time (in seconds) that the participants took to rise from a chair, walk 3 m without obstacles quickly but safely, turn around, walk back to the chair, and sit down [22]. This was measured using a stopwatch. The chair used in the test could or could not have arms (depending on availability in the participant’s house); however, the individual was not allowed to use it as support.

Four parameters were used to characterize the low PP. Low GS at baseline was determined using the cutoff point proposed by the European Working Group on Sarcopenia in Older People 2 [4], which was <0.8 m/s. A low TUG test performance at baseline was classified based on the distributions of the sample. To obtain the results, we divided the sample into tertiles and classified those in the highest tertile (longer test duration) as having low performance on the TUG test (cutoff of 11 s). We also evaluated the changes in PP between 2014 and 2019–2020. To classify the decline in PP, we defined it as a variation of ≥0.1 m/s in GS after considering previous studies [6,23] and a ≥5% in TUG time. An increase greater than or equal to 5% in the time (s) of TUG execution and a reduction greater than or equal to 0.1 m/s in GS indicated a decline in PP.

### 2.3. Outcomes

Five health outcomes were prospectively evaluated (2019–2020): (1) Negative Health Self-Perception (NHSP)—self-reported current health status was assessed through participants’ responses to the question “How do you consider your health?”, according to previous studies [24,25], with the following response options: very good, good, regular, bad, or very bad. The answers “bad” and “very bad” were considered NHSP; (2) Number of Consultations—all consultations with health professionals in the last year reported by the older adults were counted. We considered five or more consultations as a high number of consultations, based on sampling distribution; (3) Physical Disability—participants’ ability to perform activities of daily living (ADLs) was assessed using the Katz Index [26]. The instrument included six items: bathing, dressing, going to the toilet, transferring, continence, and feeding. The scoring of items was binary, with one point given for independence and none given if the individual was dependent on supervision or assistance. The participants were classified as having a functional disability if they reported needing help to perform at least one ADL; (4) Falls—at least one self-reported fall in the last year, according to previous studies [15,27,28,29,30,31,32]; and (5) Hospitalization—at least one self-reported hospitalization in the last year, according to previous studies [15,32,33], regardless of the cause of admission.

Most outcomes were self-reported by older adults or caregivers. Despite Brazil having a public health system with universal coverage, it lacks an integrated information system for medical records across different levels of care (primary, specialized, and hospital care). This limitation made it unfeasible to retrieve data on the number of health consultations, hospitalizations, and falls.

### 2.4. Covariates

A set of pPotential confounders for the association between PP and the outcomes were collected at baseline: age; sex; skin color (observed by the interviewer, considering that it is an indicator of inequality in Brazil); marital status; education level (based on years of education); socioeconomic status (according to Associação Brasileira de Empresas de Pesquisa—ABEP) [34]; current work situation; diet quality (assessed using the Diet Quality Index for the Elderly) [35]; leisure-time physical activity level (assessed by the International Physical Activity Questionnaire) [36]—those who exercised at least 150 min/week were classified as active; smoking history; alcohol consumption in the last month; multimorbidity (categorized into “up to four chronic diseases” or “five or more chronic diseases” and based on a previous study with this sample on the inequality of multimorbidity [37], taking into account 14 diseases: hypertension, diabetes, heart problems, heart failure, asthma, emphysema, ischemia or strokes, arthritis, rheumatism or arthrosis, Parkinson’s disease, loss of kidney function, high cholesterol, osteoporosis, memory problems, and cancer); depressive symptoms (according to the Geriatric Depressive Scale—GDS-10) [38,39]; polypharmacy (defined as the continuous use of five or more medications—all medicines mentioned were counted, whether or not they appeared on a medical prescription and boxes of medicines presented by the individual at the time of the interview) [40]; and body mass index (BMI), as the ratio between weight (kg) and height (m^2^). Participants were classified as low-weight, eutrophic, or overweight/obese based on age-specific cutoff points recommended by Lipschitz et al. [41] with a BMI of <22.0 kg/m^2^, 22.0–27.0 kg/m^2^, or >27.0 kg/m^2^, respectively.

### 2.5. Statistical Analysis

Pearson’s chi-square test was used to assess possible differences between the participants who did and did not complete the follow-up. Associations of low PP at baseline and changes in both tests over time with health outcomes according to sociodemographic, behavioral, and health variables were evaluated using Fisher’s exact test.

The association between PP and health outcomes was assessed using Poisson’s regression with robust variance. We chose Poisson’s regression because cross-sectional analyses with binary outcomes fit better when using Poisson’s regression than logistic regression [42]. In addition, a prevalence ratio is easier to interpret and communicate than an odds ratio. The adjustment model was defined using a directed acyclic graph (DAG) (Appendix A). The DAG was developed using the DAGitty software version 3.1, utilizing a primary set of variables to identify a minimum and sufficient number of confounders [43]. According to the DAG, the minimal and sufficient adjustment model to assess the association between PP and health outcomes comprised the following variables: economic level, diet quality, physical activity, multimorbidity, depression, polypharmacy, and BMI. Additionally, baseline GS and TUG measures were added to the adjusted model to analyze the association between changes in GS and TUG over time and health outcomes. All analyses were performed using Stata version 16.1 (College Station, TX, USA, StataCorp LP). Statistical significance was set at *p* < 0.05.

## 3. Results

Figure 1 shows the workflow of this study. In 2014, 1844 older adults were included, of which 393 (21%) were lost or refused, totaling 1451 (79%) respondents. By 30 April 2017 (the closing date of the second visit), 145 deaths had occurred (10%). Of the 900 individuals reassessed in the follow-up period (based on the previous mortality rate of this group), only 537 completed the reassessment due to the COVID-19 pandemic. This study does not include updated mortality data after 2017. Thus, considering the losses and refusals at both times and the eligibility criteria for carrying out the tests, the final sample included in the analyses consisted of 476 older adults with available information from the GS and TUG tests at baseline and follow-up (2014 and 2019–2020).

Most participants were female (65%) and were 68 ± 7 years old. Sociodemographic, behavioral, and health characteristics were similar in both assessments, except for a lower participation rate at follow-up of those aged 80 years or older and those who were widowers (Table 1). At baseline, 24% of the older adults had low GS (≤0.8 m/s), while 33% had low performance in TUG (>11 s). In addition, 68% of the participants showed a significant decline in GS and TUG performance during the study period. Approximately 6.5% of the older adults had an NHSP, 37% had a high number of consultations, 34% had physical disabilities, 27% had at least one fall during the previous year, and 13% had been hospitalized in the previous year (Figure 2).

Appendix A summarize the association between PP measures and outcomes according to the covariates. Low PP (low GS and/or high TUG time) at baseline was associated with older age, a lower level of education, no polypharmacy (*p* < 0.001 for all, GS and TUG), being female (*p* = 0.001 for GS and *p* = 0.003 for TUG), lower socioeconomic status (*p* = 0.001 for GS), multimorbidity (*p* = 0.001 for GS and *p* < 0.001 for TUG), depression (*p* = 0.001 for GS and *p* = 0.002 for TUG), being unemployed (*p* = 0.002 for GS and *p* = 0.023 for TUG), and being a widower (*p* = 0.008 for GS and *p* = 0.002 for TUG). However, PP decline was not associated with any socioeconomic, demographic, behavioral, or health characteristics.

NHSP was associated with depressive symptoms (*p* < 0.001), physical inactivity (*p* = 0.001), multimorbidity (*p* = 0.008), and absence of polypharmacy (*p* = 0.029). A high number of consultations was associated with lower education and socioeconomic status (*p* = 0.001 and 0.006, respectively), polypharmacy (*p* = 0.007), female sex (*p* = 0.011), and multimorbidity (*p* = 0.026). Physical disability was associated with older age, multimorbidity (*p* < 0.001), male sex (*p* = 0.001), lower economic level (*p* = 0.005), depression (*p* = 0.016), no polypharmacy (*p* = 0.017), lower education level (*p* = 0.027), and no alcohol consumption in the previous month (*p* = 0.046). Falls were associated with female sex (*p* < 0.001) and depressive symptoms (*p* = 0.036). Hospitalization was not associated with the covariates.

Associations between PP and health outcomes are presented in Table 2 and Table 3. Crude analyses indicated an association between the GS and TUG test results at baseline and NHSP (*p* = 0.007 and *p* = 0.003, respectively), physical disability (*p* < 0.001 for both), and falls (*p* = 0.032 and *p* = 0.002, respectively). However, in the adjusted model, PP tests were not associated with NHSP or a high number of consultations. A low PP on the TUG test at baseline was associated with a higher risk of physical disability (PR = 1.60, 95%CI 1.24; 2.06, *p* < 0.001) and a 57% higher risk of falls (PR = 1.57, 95%CI 1.13; 2.18, *p* = 0.007). A low GS at baseline was associated with a higher risk of physical disability (PR = 1.56, 95%CI 1.18, 2.06, *p* = 0.002). The decline in GS over time was the only measure associated with hospitalization (PR = 1.86, 95%CI 1.05, 3.31, *p* = 0.033). Although no association was found between the type of medical appointment and physical performance tests, the analyses are shown in Appendix A.

## 4. Discussion

Our findings suggest that PP is associated with disability, falls, and hospitalization after five to six years among Brazilian community-dwelling older adults, residents of a medium-sized city, in the state with the highest percentage of older people in the country [2]. However, PP tests were not associated with a NHSP or the number of consultations with health professionals. Low PP is widely recognized as a risk factor for adverse health outcomes, including disability, falls, and hospitalization among community-dwelling older adults in Brazil [12], United States [6,15], Australia [27], Japan [44,45,46,47], Korea [48], and European countries [49,50], in agreement with our findings. PP integrates different physiological systems (cardiac, respiratory, and neuromuscular) in coordinated and efficient movements, reflecting the function of the entire body, and is objectively measured in relation to the individual’s ability to move [51]. Thus, it has been considered as a “sixth vital sign” [3].

In our study, PP assessed by both the GS and TUG at baseline increased the risk of physical disability, with similar practical results (PR = 1.60 and PR = 1.57, respectively), which coincides with previous studies on community-dwelling older adults [44,45,46,47,49,50]. Impairments in performing ADLs can be understood as a consequence of a decline in PP. Abe et al. (2019) [44] showed that GS was a significant independent predictor of incident disability over a 4.4-year period and that physical activity did not mediate this association.

Our results showed that only low performance, as assessed by the TUG test time at baseline, was associated with a higher prevalence of falls. TUG is the most commonly used tool in studies assessing fall risk, precisely because it evaluates, in addition to walking ability, other dimensions such as balance and strength [22], while GS has shown inconsistent results for predicting falls [6,52,53]. The Canadian Longitudinal Study on Aging verified that neither of the commonly used PP tests (TUG and GS) achieved acceptable accuracy in identifying individuals with at least one fall at follow-up (18 months) [52]. This is in agreement with previous studies in which the values of sensitivity and specificity ranged from weak to moderate for the TUG and 4 m GS tests [28,29,30,33,54,55]. However, other studies have shown consistent results regarding the association of both the TUG and GS tests with the risk of falls among community-dwelling older adults worldwide, in addition to being cost-effective measures for application in different contexts [29,30,31,32,53,56,57]. The divergence of results across studies may be due, in part, to variability between protocols for performing the walking tests and the cutoff points used for fall risk detection, especially in the TUG test. Falls are a significant problem among older adults and should be the target of prevention through investment in public programs that demonstrate that a reduction in the risk of falls is possible with an improvement in PP [58].

Decline in GS from 2014 to 2019–2020 was a predictor of increased risk of hospitalizations (PR = 1.86, 95%CI = 1.05; 3.31) among community-dwelling older Brazilians. The assessment of PP decline (instead of the isolated use of PP tests at baseline) can identify individuals who were previously above the low performance threshold, but over time accumulated risk factors that contributed to both PP decline and the higher risk of hospitalizations, such as multimorbidity, polypharmacy, and depressive symptoms, in addition to socioeconomic and behavioral factors, as previously investigated in our sample [9]. A recent study found that GS decline was the only determinant associated with hospital readmission (OR = 0.35, 95%CI = 0.16–0.79) [27]. This is consistent with the catabolic model, which enhances age-related loss of muscle mass and function [59,60]. Given that hospitalizations represent a high cost to the health system and can be prevented, it is essential that health professionals and managers include PP tests in the evaluation and monitoring of the health of older adults over time [12].

In contrast, low PP was not associated with an NHSP or with the number of consultations with health professionals in our sample. Interestingly, older people with low PP may also have risk factors such as physical inactivity, multimorbidity, and depressive symptoms, which, in our study, showed an association with NHSP. Brazilian studies have suggested that a TUG time of >10 s is associated with NHSP among community-dwelling older adults [24,25]. This divergence is probably related to the design of the study (cross-sectional) and/or variables included in the analysis model, given that our crude analyses also identified an association of PP tests at baseline with NHSP; however, after including a set of potential confounders, this association disappeared. The number of consultations might not be a good parameter to assess unnecessary health expenses for heartsink patients, for example, because it is expected that individuals with low PP are the target of more interventions, including consultations with a multidisciplinary team [61,62,63]. This relationship could be better explored in studies conducted in developed countries, whose health system has an integrated information system on consultations, procedures, and access to specialized services, which has not yet occurred in Brazil due to the lack of communication between primary care health services and specialized services.

It is necessary to point out that PP measures and the different outcomes analyzed may be interdependent. A low PP increases the risk of these outcomes; however, once they occur, it also negatively impacts PP and the likelihood of other outcomes occurring. This assumption was also confirmed in previous studies, which pointed out that individuals who had experienced previous hospitalizations were more likely to exhibit GS decline and had higher odds of new ADL limitations [64,65]. Considering that PP improvement can prevent and/or reverse adverse health outcomes, timely interventions related to physical exercise and nutritional approaches to maintain functionality, guarantee independence, and improve quality of life should be directed at this age group [66].

Furthermore, few studies of PP among community-dwelling older adults have simultaneously evaluated multiple outcomes [12,32,33,67]. For example, Batko-Szwaczka et al. (2020) found that the TUG test was the only independent measure able to predict the occurrence of a combined outcome (falls, hospitalization, and mortality) within one year (OR = 1.22; 95%CI 1.07–1.40, *p* = 0.003) [33]. Welch et al. (2016) observed that a 1 s increase in GS was associated with a 26% higher risk of falls (RR = 1.26, 95%CI = 1.10–1.45) [32]. Nevertheless, the GS test was not associated with a history of injuries or hospitalization related to falls.

To the best of our knowledge, this is the first study to address low PP related to multiple adverse health outcomes simultaneously in community-dwelling older adults using a combination of PP measures at a single point (cutoff for low threshold) and long-term changes (according to clinically significant changes in physical tests). As a strength, the present study has a longitudinal design and a robust statistical model of adjustment to evaluate the association between PP and multiple outcomes, including health indicators of the older people that are still little investigated, such as health self-perception and number of health consultations. In addition, the use of two widely recommended PP tests allows comparison with populations across different countries. This study provides data from a representative sample of community-dwelling older people who live in the urban area of the Brazilian state with the highest percentage of the older population (around 20%) [2], filling a gap related to the scarcity of research in low- and middle-income countries in Latin America.

This study has certain limitations. First, we assumed that not all participants experienced an outcome at baseline that corresponded to a real situation, thus suggesting a potential attenuation in the effect measures found. Second, although the analyses were adjusted for a considerable number of potential confounders, the long time between baseline and reassessment increases the chance that other factors not considered here may have affected the investigated associations. Third, the interruption of follow-ups in 2020 due to the COVID-19 pandemic reduced the participation of older people. However, this did not affect the detection of significant associations in the samples. The number of falls and reasons for fall recurrence were not analyzed, which may also be a limitation in the interpretation of these data. Furthermore, the criteria used to determine a significant decline in the GS can be troublesome and lead to classification errors. It may be easier to identify variations in the fastest, but not the slowest, individuals. However, it should be noted that our study used the values described for significant clinical changes in PP [6,23]. Finally, this sample may not be representative of the whole country, given social inequalities, differences in technological density, and access to health services between Brazilian states, which may impact the outcomes analyzed, requiring caution regarding the generalization of the data.

Our findings suggest that different measures of PP (evaluated at a single point and over six years) showed an association with important preventable outcomes among older adults. Therefore, more longitudinal studies are needed, especially in Latin America, to evaluate the role of PP in predicting multiple adverse health outcomes in the short-, medium-, and long-term. We also observed that while one PP test was associated with a certain outcome, another may not have shown this association. Thus, they are complementary as they assess different dimensions, especially when evaluating samples of younger and older people. Interestingly, one study found little overlap in the trajectories of different physical tests among individuals aged 60–70 years [18]. Likewise, future studies with representative samples of community-dwelling older adults in different scenarios assessing health self-perceptions and the number of consultations with health professionals in the context of PP are necessary for a better understanding. Moreover, as well as exploring other outcomes such as fall-related fractures, the need for rehabilitation and long-term care needs to be explored to direct actions that impact quality of life.

## 5. Conclusions

This study showed that low PP was associated with physical disability, falls, and hospitalizations but not with a NHSP or a high number of consultations among Brazilian community-dwelling older adults over a period of five to six years. The combination of physical tests (GS and TUG) and different measures (evaluated at a single point and over six years) reinforce the importance of including PP in the yearly health assessment of the older adults and monitoring this indicator to target cost-effective measures in a timely manner, especially in low- and middle-income countries such as Brazil, in order to prevent negative health outcomes and promote equitable aging for all.

## Figures and Tables

**Figure 1 ijerph-21-00319-f001:**
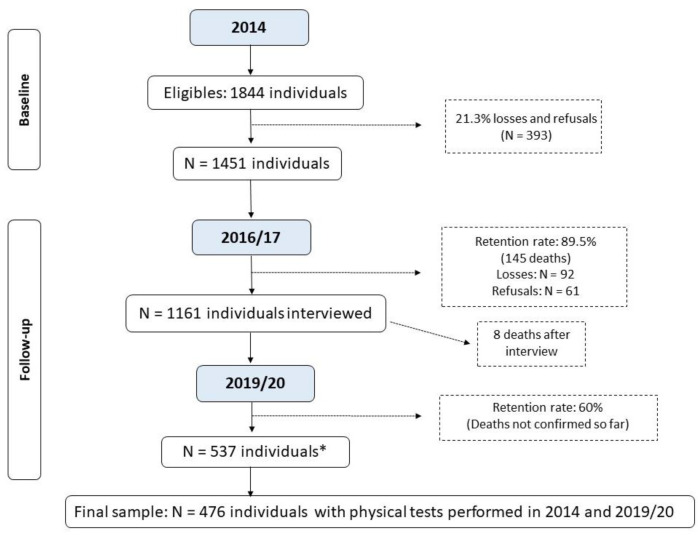
Flowchart of the Longitudinal Study on Elderly Health: continuing the “COMO VAI?” study. * Number of living elderly located and interviewed before the study was interrupted due to the COVID-19 pandemic; until April/2023, the deaths that occurred were not verified by the epidemiological surveillance of Pelotas due to the duration of the pandemic.

**Figure 2 ijerph-21-00319-f002:**
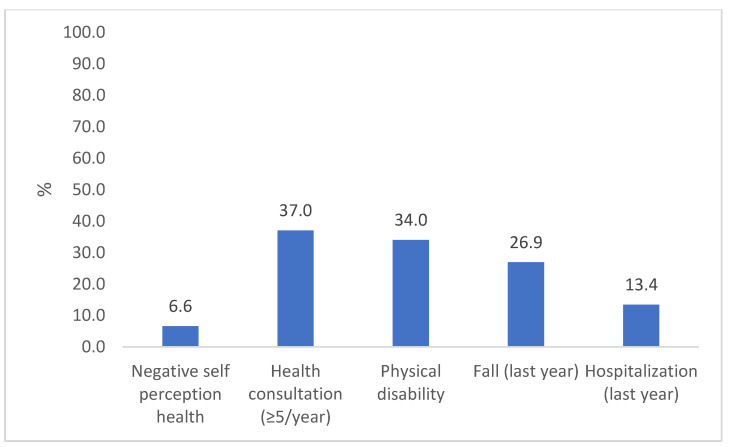
Prevalence of adverse health outcomes in the longitudinal study “COMO VAI?”. N = 476, Pelotas, 2019–2020.

**Table 1 ijerph-21-00319-t001:** Sample description according to demographic, socioeconomic, behavioral, and health-related characteristics in the 2014 and 2019 years of the COMO VAI? study.

	Complete Sample in 2014		Study Sample in 2019	
Variables	N = 1451	% (95%CI) ^a^	N = 476	% (95%CI) ^a^
Sex				
Female	914	63.0 (60.5; 65.4)	310	65.1 (60.7; 69.3)
Male	537	37.0 (34.6; 39.5)	166	34.9 (30.7; 39.3)
Age (completed years)				
60–69	756	52.3 (49.7; 54.9)	289	60.7 (56.2; 65.0)
70–79	460	31.8 (29.5; 34.3)	151	31.7 (27.7; 36.1)
80+	230	15.9 (14.1; 17.9)	36	7.6 (5.5; 10.3)
Skin color				
White	1211	83.7 (81.7; 85.6)	389	81.7 (78.0; 85.0)
Other than white	236	16.3 (14.5; 18.3)	87	18.3 (15.0; 22.0)
Marital status				
Married/with a partner	763	52.7 (50.2; 55.3)	286	60.1 (55.6; 64.4)
Without a partner/separated	225	15.6 (13.8; 17.6)	72	15.1 (12.2; 18.6)
Widow(er)	459	31.7 (29.4; 34.2)	118	24.8 (21.1; 28.9)
Economic level ^b^				
A/B	483	35.2 (32.7; 37.8)	161	35.4 (31.1; 39.9)
C	720	52.5 (49.8; 55.1)	248	54.5 (49.9; 59.0)
D/E	169	12.3 (10.7; 14.2)	46	10.1 (7.7; 13.2)
Education level (completed years)				
None	196	13.6 (12.0; 15.5)	54	11.4 (8.8; 14.6)
1–7	782	54.4 (51.8; 57.0)	270	56.8 (52.3; 61.2)
≥8	459	32.0 (29.6; 34.4)	151	31.8 (27.7; 36.1)
Current work situation				
No (unemployed)	1084	80.4 (78.2; 82.4)	344	77.0 (72.8; 80.6)
Yes (employed)	264	19.6 (17.6; 21.8)	103	23.0 (19.4; 27.2)
Diet quality ^c^				
Low	481	33.7 (31.3; 36.2)	143	30.2 (26.3; 34.6)
Average	534	37.5 (35.0; 40.0)	177	37.5 (33.2; 41.9)
High	411	28.8 (26.5; 31.2)	153	32.3 (28.3; 36.7)
Leisure-time physical activity ^d^				
≤150 min/week	1133	81.5 (79.3; 83.4)	378	80.4 (76.6; 83.8)
>150 min/week	258	18.5 (16.6; 20.7)	92	19.6 (16.2; 23.4)
Smoking				
Not a smoker	781	54.0 (51.4; 56.6)	262	55.0 (50.5; 59.5)
Smoker	182	12.6 (11.0; 14.4)	58	12.2 (9.5; 15.5)
Former smoker	483	33.4 (31.0; 35.9)	156	32.8 (28.7; 37.1)
Alcohol consumption ^e^				
No	1138	78.8 (76.6; 80.8)	355	74.6 (70.5; 78.3)
Yes	307	21.2 (19.2; 23.4)	121	25.4 (21.7; 29.5)
Multimorbidity				
Up to 4 diseases	473	35.3 (32.8; 37.9)	175	37.6 (33.3; 42.1)
5 or more diseases	866	64.7 (62.1; 67.2)	291	62.4 (57.9; 66.7)
Depression ^f^				
No	1182	84.8 (82.8; 86.6)	408	86.4 (83.0; 89.3)
Yes	212	15.2 (13.4; 17.2)	64	13.6 (10.7; 17.0)
Polypharmacy ^g^				
0–4 medications	513	35.6 (33.1; 38.1)	150	31.5 (27.5; 35.8)
≥5 medications	929	64.4 (61.9; 66.9)	326	68.5 (64.2; 72.5)
BMI ^h^				
<22.0 kg/m^2^	126	9.2 (7.8; 10.9)	26	5.5 (3.8; 8.0)
22.0–27.0 kg/m^2^	471	34.5 (32.1; 37.1)	152	32.1 (28.1; 36.4)
>27.0 kg/m^2^	767	56.3 (53.6; 58.8)	295	62.4 (57.9; 66.6)
Physical Performance Tests				
Gait Speed (GS) in 2014 ^i^				
Normal (>0.8 m/s)	994	76.2 (73.8; 78.4)	385	82.1 (78.3; 85.3)
Low (≤0.8 m/s)	311	23.8 (21.6; 26.2)	84	17.9 (14.7; 21.7)
Change in GS (2014–2019) ^j^				
Stable–Improvement	-	-	152	31.9 (27.9; 36.3)
Decline	-	-	324	68.1 (63.7; 72.1)
Timed Up and Go (TUG) in 2014 ^k^				
Normal (≤11 s)	881	67.3 (64.7; 69.7)	340	72.2 (68.0; 76.1)
Low performance (>11 s)	429	32.7 (30.3; 35.3)	131	27.8 (23.9; 32.0)
Change in TUG (2014–2019) ^j^				
Stable–Improvement	-	-	150	31.9 (27.8; 36.2)
Worsening	-	-	321	68.1 (63.8; 72.2)

^a^ Pearson’s chi-square test; ^b^ according to *Associação Brasileira de Empresas de Pesquisa* (ABEP): category A/B indicates higher socioeconomic status; ^c^ assessed using the Diet Quality Index for the Elderly (*Índice de Qualidade da dieta do idoso*—IDQ-I); ^d^ assessed by the International Physical Activity Questionnaire (IPAQ); ^e^ alcohol consumption in the last month; ^f^ according to the Geriatric Depressive Scale (GDS-10); ^g^ continuous use of five or more medications; ^h^ cutoff points recommended by Lipschitz et al.; ^i^ cutoff point recommended by the European Working Group on Sarcopenia in Older People 2 (2019); ^j^ clinically relevant changes were defined as a variation of ≥0.1 m/s in GS and ≥5% in TUG time. ^k^ cutoff points according to sample distribution: lower tertile defines low performance.

**Table 2 ijerph-21-00319-t002:** Baseline Gait Speed (GS) and Timed Up and Go (TUG) tests and the change in both tests from 2014 to 2019 in older adults who experienced negative self-perception health and high number of health consultations.

Variables	Negative Health Self-Perception ^a^ N = 32		Health Consultations (≥5 Per Year) N = 156	
	Crude *p*-Value PR (95%CI)	Adjusted *p*-Value ^b^ PR (95%CI)	Crude *p*-Value PR (95%CI)	Adjusted *p*-Value ^b^ PR (95%CI)
GS (m/s) in 2014 ^c^ (n = 1305)	0.007	0.327	0.272	0.261
Normal (>0.8 m/s) (n = 994)	1.00	1.00	1.00	1.00
Low (≤0.8 m/s) (n = 311)	2.65 (1.31; 5.37)	1.52 (0.66; 3.52)	1.19 (0.88; 1.61)	1.21 (0.87; 1.69)
Change in GS (2014–2019) ^d^ (n = 476)	0.791	0.488	0.910	0.718
Stable–Improvement (n = 152)	1.00	1.00	1.00	0.820
Decline (n = 324)	1.11 (0.52; 2.36)	1.28 (0.63; 2.61)	0.98 (0.75; 1.29)	1.04 (0.77; 1.39)
TUG (s) in 2014 ^e^ (n = 1433)	0.003	0.085	0.121	0.243
Normal (≤11 s) (n = 1231)	1.00	1.00	1.00	1.00
Low performance (>11 s) (n = 202)	2.77 (1.41; 5.44)	1.83 (0.92; 3.64)	1.23 (0.95; 1.60)	1.18 (0.89; 1.57)
Change in TUG (2014–2019) ^d^ (n = 471)	0.653	0.836	0.699	0.796
Stable–Improvement (n = 150)	1.00	1.00	1.00	1.00
Worsening (n = 321)	0.85 (0.42; 1.73)	0.93 (0.46; 1.88)	0.95 (0.73; 1.24)	1.04 (0.77; 1.40)

^a^ participants who answered “poor” or “very poor” to the question “how do you rate your health?”. ^b^
*p*-value using Poisson’s regression adjusting for the following variables: economic level, diet quality, physical activity, multimorbidity, depression, polypharmacy, and BMI. ^c^ cutoff point recommended by the European Working Group on Sarcopenia in Older People 2 (2019). ^d^ clinically relevant changes were defined as a variation of ≥0.1 m/s in GS and ≥5% in TUG time. ^e^ cutoff points according to sample distribution: lower tertile defines low performance.

**Table 3 ijerph-21-00319-t003:** Baseline Gait Speed (GS) and Timed Up and Go (TUG) tests and the change in both tests from 2014 to 2019 in older adults who experienced physical disability, falls, and hospitalization in the last year.

Variables	Physical Disability ^a^ N = 162		Falls (Last Year) N = 130		Hospitalization (Last Year) N = 65	
	Crude *p*-Value PR (95%CI)	Adjusted *p*-Value ^b^ PR (95%CI)	Crude *p*-Value PR (95%CI)	Adjusted *p*-Value ^b^ PR (95%CI)	Crude *p*-Value PR (95%CI)	Adjusted *p*-Value ^b^ PR (95%CI)
GS (m/s) in 2014 ^c^ (n = 1305)	<0.001	0.002	0.032	0.104	0.914	0.980
Normal (>0.8 m/s) (n = 994)	1.00	1.00	1.00	1.00	1.00	1.00
Low (≤0.8 m/s) (n = 311)	1.92 (1.50; 2.45)	1.56 (1.18; 2.06)	1.44 (1.03; 2.02)	1.37 (0.94; 2.01)	0.97 (0.53; 1.77)	0.99 (0.53; 1.87)
Change in GS (2014–2019) ^d^ (n = 476)	0.900	0.326	0.072	0.221	0.341	0.033
Stable–Improvement (n = 152)	1.00	1.00	1.00	1.00	1.00	1.00
Decline (n = 324)	0.98 (0.75; 1.29)	1.15 (0.87; 1.53)	0.76 (0.56; 1.03)	0.81 (0.58; 1.14)	1.29 (0.76; 2.17)	1.86 (1.05; 3.31)
TUG (s) in 2014 ^e^ (n = 1433)	<0.001	<0.001	0.002	0.007	0.341	0.303
Normal (≤11 s) (n = 1231)	1.00	1.00	1.00	1.00	1.00	1.00
Low performance (>11 s) (n = 202)	1.96 (1.54; 2.49)	1.60 (1.24; 2.06)	1.59 (1.18; 2.15)	1.57 (1.13; 2.18)	1.26 (0.78; 2.05)	1.32 (0.78; 2.26)
Change in TUG (2014–2019) ^d^ (n = 471)	0.651	0.065	0.254	0.717	0.682	0.303
Stable–Improvement (n = 150)	1.00	1.00	1.00	1.00	1.00	1.00
Worsening (n = 321)	1.07 (0.81; 1.41)	1.30 (0.98; 1.71)	0.84 (0.61; 1.14)	0.94 (0.65; 1.34)	1.11 (0.67; 1.83)	1.34 (0.77; 2.32)

^a^ according to Katz Index. ^b^ *p*-value using Poisson’s regression adjusting for the following variables: economic level, diet quality, physical activity, multimorbidity, depression, polypharmacy, and nutritional status. ^c^ cutoff point recommended by the European Working Group on Sarcopenia in Older People 2 (2019). ^d^ clinically relevant changes were defined as a variation of ≥0.1 m/s in GS and ≥5% in TUG time. ^e^ cutoff points according to sample distribution: lower tertile defines low performance.

## Data Availability

The data used in this study can be obtained upon request from the corresponding author.

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
