# Peer review of "Low Physical Performance Could Be Associated with Adverse Health Outcomes over Time: Results from a Cohort of Older Adults"

_ijerph, 2024, doi:10.3390/ijerph21030319_

Round 1

Reviewer 1 Report

Comments and Suggestions for Authors

Thank you for inviting me to review the manuscript entitled “Low physical performance could be associated with adverse health outcomes over time: results from a cohort of older adults” by Gomes et al.

The study addresses health outcomes in relation to physical performance in the elderly. This topic is appropriately addressed and of interest to the field. It presents originality by covering several unexplored aspects of the problem, such as performance indicators in elderly people. On the other hand, the authors could consider adding physiological and psychological indicators related to aging that can affect performance, such as testosterone levels, and also enhancing the introduction section.

The methodology section also benefits from improvements: the sample estimation method should be added, graphs should be added to show the differences and the validity and reliability of the tests.

The discussion should be focused on the factors of decrease or increase, improvement or non-improvement, and related mechanisms should be mentioned. A chart representation would be more appropriate

The references are appropriate, however they could be updated.

Reviewer 2 Report

Comments and Suggestions for Authors

I congratulate the authors for the study. Long-term studies make significant contributions to the literature and society. The current study appears to be generally well-constructed and has an appropriate methodology. In addition, the parts that I think will contribute to the enrichment of the study are highlighted below.

Could you introduce physical performance in a different paragraph. The relationship between physical performance and health and the importance of physical activity in older people should be further explained.

Data on old age in Brazil, its impact on the health system, and the state's policies towards older people should be mentioned.

In the last paragraph of the introduction section, the possible contribution of the result of this study to the literature, society, and the policies of states (in particular, the Brazilian state) should be emphasized.

In the discussion section, the findings obtained should be highlighted as the Brazilian example and discussed with data obtained from other countries, highlighting similarities or differences.

Reviewer 3 Report

Comments and Suggestions for Authors

This is a very interesting article presenting longitudinal data, but some points need a major revision:

The first sentences in the introduction are very general and not related to the topic under examination.

Please also discuss the importance and the various effects in different domains of frailty as presented in https://doi.org/10.14283/jfa.2022.57

Please support with more references the hypotheses especially in the introduction (and present them more clearly in the text).

Why is over 60 years of age a cutoff point? Usually 65 is used.

Sample size and sample recruitment must be described in this text.

Why are older adults in Brazil 'a special population'? It is not clear why?

How is polypharmacy defined? Why over 5 medications is considered to be the case?

In table 3 what does crude mean?

Comments on the Quality of English Language

Moderate English language editing.
